# Rotational Barriers in N-Benzhydrylformamides: An NMR and DFT Study

**DOI:** 10.3390/molecules28020535

**Published:** 2023-01-05

**Authors:** Madina Zh. Sadvakassova, Andrei I. Khlebnikov, Abdigali A. Bakibaev, Oleg A. Kotelnikov, Rakhmetulla Sh. Erkassov, Madeniyet A. Yelubay, Manar A. Issabayeva

**Affiliations:** 1Department of Chemistry, L.N. Gumilyov Eurasian National University, 010008 Astana, Kazakhstan; 2Kizhner Research Center, Tomsk Polytechnic University, 634050 Tomsk, Russia; 3Faculty of Chemistry, National Research Tomsk State University, 634050 Tomsk, Russia; 4Department of Chemistry and Chemical Technology, Toraighyrov University, 140008 Pavlodar, Kazakhstan

**Keywords:** N-benzhydrylformamide derivatives, rotational barriers, dynamic NMR, DFT calculations

## Abstract

N-Benzhydrylformamides are pharmacologically active compounds with anticonvulsant, enzyme-inducing, antihypoxic, and other types of biological activity. The conformational behavior of benzhydrylformamides is determined to a great extent by the presence of substituents at the nitrogen atom and in the *ortho*-position(s) of the diphenylmethane moiety. Particularly, the NMR spectra of these compounds often contain two sets of signals originating from different orientations of the formyl group. With the use of the dynamic NMR method and DFT calculations, we investigated the internal rotations of aromatic and formyl fragments and estimated the corresponding rotational barriers in N-benzhydrylformamide (BHFA), N-methyl-N-benzhydrylformamide (BHFA-NMe), and in a series of *ortho*-halogen-substituted N-benzhydrylformamides. It was found that the DFT method at M06-2X/6-311+G* level of theory satisfactorily reproduces the experimental barrier $\Delta G_{298}^{\neq }$(Formyl) of the formyl group rotation in BHFA-NMe. In BHFA, BHFA-NMe, and in the *ortho*-halogen derivatives, the calculated $\Delta G_{298}^{\neq }$(Formyl) values are close to each other and lie within 20–23 kcal/mol. On the other hand, the *ortho*-substituents significantly hinder the rotation of aryl fragment with $\Delta G_{298}^{\neq }$(Aryl) values varying from 2.5 kcal/mol in BHFA to 9.8 kcal/mol in *ortho*-iodo-N-benzhydrylformamide.

## 1. Introduction

The presence of a benzhydryl substituent is characteristic of many biologically active compounds. For example, benzhydrylformamides exhibit anticonvulsive [1,2,3], antiviral, antibacterial [4], and antihypoxic properties [5]. Other types of biological activity of benzhydrylformamides and their structural features were also investigated [5,6]. Along with characteristics of substituents, the conformational behavior of these compounds often plays a decisive role in the manifestation of biological activity, and the slightest structural changes affect the chemical properties of benzhydrylformamides [7]. In the present work, we studied the effect of substituents of the same type in the benzhydryl fragment on the conformational behavior and electronic structure of some pharmacologically active benzhydrylformamides. The objects of the study were N-benzhydrylformamide (BHFA), its *ortho*-fluoro, *ortho*-chloro-, *ortho*-bromo- and *ortho*-iodo derivatives (BHFA-oF, BHFA-oCl, BHFA-oBr and BHFA-oI, respectively), as well as N-benzhydryl-N-methylformamide (BHFA-NMe) (Figure 1).

We studied the conformational equilibrium of the BHFA-NMe compound in a dimethyl sulfoxide (DMSO) solution by dynamic NMR, observing the coalescence of N-methyl group proton signals with increasing temperature, and also determined the internal rotation barriers of the compounds presented in Figure 1 using quantum chemical calculations by the DFT method.

The choice of the studied benzhydrylformamides (Figure 1), which were previously synthesized according to the method [7], was based on the fact that the substituents in the *ortho* position and at the nitrogen atom obviously should cause the most pronounced conformational changes compared to the parent molecule BHFA.

## 2. Results and Discussion

### 2.1. Investigation of the Internal Rotation by NMR

Using the NMR method, it is possible to determine the barrier of internal rotation in a molecule if the rate of mutual transformation of conformers (isomers) is not too high, and the nuclei exchange their chemical environment at a rate slower than the NMR time scale [8]. In this case, an increase in temperature (T) leads to a gradual convergence of distinct NMR signals of individual conformers and to their merging into one signal at a certain temperature T_c_ (the coalescence temperature).

Internal rotation around the C-N bond in the HC(O)-N group of formamides is quite slow due to the conjugation between the nitrogen atom and the carbonyl group [9,10]. Therefore, the signals of both conformers are observed separately in the ^1^H NMR spectra at room temperature. Thus, in the spectrum of BHFA-NMe (Figure 1), there are two signals of methyl groups at 2.64 and 2.71 ppm. Further, the paired signals of CH methine protons at 6.23 and 6.68 ppm, as well as formyl protons at 8.28 and 8.36 ppm, are present, which are attributed to two different conformers.

With recurring registration of the ^1^H NMR spectrum with increasing temperature, we observed that the signals of methyl protons were converged and merged into a single peak at 110 °C and higher (Figure 2).

The found coalescence temperature T_c_ = 110 °C (383 K) makes it possible to calculate the rotational barrier between two conformers in which the carbonyl group is in *syn*- or *anti*-orientation with respect to the benzhydryl group of the BHFA-NMe molecule (Figure 3).

In addition to the value of T_c_, for the calculation of the rotational barrier, it is necessary to determine the distance Δν between the corresponding individual conformer signals at a low temperature [8]. From the ^1^H NMR spectrum recorded at 20 °C (Figure 1), it can be found that the Δν value for the methyl group signals equals 27.2 Hz. Calculation of the barrier ΔG^≠^ using Equation (1) leads to a value of 19.5 kcal/mol. This value is comparable with the rotational barriers in other molecules containing a formamide fragment [9,10].

### 2.2. Calculation of the Internal Rotational Barriers by the DFT Method

#### 2.2.1. Calculation of the Rotational Barriers in N-Benzhydryl-N-Methylformamide

To evaluate the possibilities of the DFT quantum chemical method to reliably evaluate the rotational barriers in substituted benzhydrylformamides, we compared the ΔG^≠^ value obtained experimentally by the NMR method (see above) with the corresponding barrier calculated for the *syn*-*anti* conformational transition of the formyl moiety in compound BHFA-NMe. We applied M06-2X functional suitable for calculations of thermodynamic properties of organic compounds [11]. The used basis set 6-311+G* of triple-zeta quality contained diffuse functions and one set of polarization functions on heavy atoms to adequately account for interactions of lone electron pairs. The IEFPCM solvation model [12] was applied to evaluate the bulk effect of dimethylsulfoxide, in spite of a high polarity of the solvent. Using the explicit solvation model would be very complicated due to a variety of possible molecular ensembles formed between BHFA-NMe and dimethylsulfoxide. On the other hand, recording NMR spectra of benzhyldylformamides in less polar solvents was difficult because of low solubility.

Considering that the BHFA-NMe molecule has several internal rotational degrees of freedom and is, thus, a conformationally flexible compound, we preliminarily performed a conformational search by the molecular mechanics method using the VeraChem software. This search retrieved 66 representative conformations. Further, the first ten low-energy conformations were optimized by the DFT method. Their Gibbs energies are presented in Appendix A; the corresponding Gaussian output files are archived in Appendix A. The lowest-energy conformation obtained after the DFT optimization is shown in Figure 4. It has a *syn*-arrangement of the carbonyl and benzhydryl groups: the torsion angle O=C-N-CH is 0.3°. Two benzene rings are non-coplanar: the torsion angles N-CH-C_Ar_=C_Ar_ formed by them are 80.2° and −2.0°. This conformation was used as a starting point for one-dimensional relaxed scans of the potential energy surface (PES). In one of the scans, the torsion angle corresponding to the formyl fragment rotation was varied. In the other scan, one of the phenyl groups within the benzhydryl substituent was rotated. It was found that the position of another phenyl group and the mutual rotation of the benzhydryl and formamide moieties are strongly correlated with the changes in two torsion angles mentioned above. The resulting energy profiles are shown in Figure 5.

With the internal rotation of the formyl group, two maxima are observed on the energy profile (Figure 5a) over the full period (360°). The curve corresponding to the rotation of the benzene ring (Figure 5b) has four maxima approximately equal in height, which is consistent with the symmetry of the unsubstituted phenyl group. Due to the periodicity of energy profiles, a mutual transition between any two conformations (minima on the obtained curves) is possible by overcoming the energy barrier not exceeding the penultimate maximum found in the full period of internal rotation, and characteristics of this maximum should be compared with experimentally found ΔG^≠^ value (Section 2.1). The positions of the maxima, penultimate in height on the energy profiles, were refined by searching for the corresponding saddle points of the PES, followed by an analysis of normal vibrations and the calculation of thermodynamic functions at the coalescence temperature (T_c_ = 383 K). Using Gibbs energies of the low-energy conformer and the transition state (saddle point), we estimated the barrier height $\Delta G_{383}^{\neq }$(Formyl) = 23.1 kcal/mol. This value satisfactorily corresponds to the barrier of 19.5 kcal/mol determined by us experimentally (Section 2.1). Some difference may be due to the effects of specific solvation not taken into account within the IEFPCM model. The calculation of thermodynamic functions by the DFT method at 298.15 K leads to the value $\Delta G_{298}^{\neq }$(Formyl) = 22.7 kcal/mol.

It should be noted that the conformation with *anti*-orientation of the carbonyl and benzhydryl groups, in which the torsion angle O=C-N-CH equals 180.3°, differs in Gibbs energy from the low-energy *syn* conformation by only 0.15 kcal/mol, in accordance with the presence of two substituents at the formamide nitrogen atom. This result agrees with the observation that after a long-term storage of BHFA-NMe solution in DMSO-d_6_ at 20 °C, the ratio of the integral intensities of *syn*- and *anti*-methyl group singlets in the ^1^H NMR spectrum was not changed, despite the relatively low rotational barrier. The *syn*/*anti* ratio of BHFA-NMe conformers calculated from their Gibbs energies is equal to 1.3/1 in a satisfactory agreement with relative integral intensities of the NMR signals at 2.64 and 2.72 ppm (1.7/1 at 20 °C, Figure 1).

For the rotation of phenyl group in compound BHFA-NMe, the calculated barrier $\Delta G_{298}^{\neq }$(Aryl) equals 3.06 kcal/mol.

#### 2.2.2. Calculation of Rotational Barriers in N-Benzhydrylformamide and Its *ortho*-Substituted Derivatives

The results presented above show that the DFT method can be applied to evaluate rotational barriers in benzhydrylformamides. We performed quantum chemical calculations for N-benzhydrylformamide and its derivatives containing a halogen atom in the *ortho* position of one of the benzene rings. Compounds with *ortho*-substituents in the aromatic ring are of the greatest interest in terms of studying steric interactions and their influence on the conformational behavior of molecules. The 6-311+G* basis set is not defined for bromine and iodine; hence, for these atoms, we applied LANL2DZ basis with effective core potential, which was successfully used in conjunction with Pople basis sets (see, e.g., [13]).

By analogy with compound BHFA-NMe, we performed a molecular mechanics search for conformations for BHFA, BHFA-oF, BHFA-oCl, BHFA-oBr, and BHFA-oI using the VeraChem software. For ten low-energy conformations of each compound, the geometry was optimized by the DFT method. Then, for the conformation with the lowest energy value, which, in all cases, had the *syn*-orientation of the carbonyl group and the benzhydryl fragment, one-dimensional PES scans were performed, varying the torsional angle N-CH-C_Ar_=C_Ar_ (involving the benzene ring with an *ortho* substituent), or torsion angle O=C-N-CH. Energy profiles found on the scanning are shown in Figure 6 and Figure 7.

On each of the obtained profiles, the maximum penultimate in height was found, for which the search for a saddle point (transition state) on the PES was performed. The thermodynamic characteristics of the lowest-energy conformers and transition states made it possible to calculate the barriers of internal rotation of the formyl and aryl groups ($\Delta G_{298}^{\neq }$(Formyl) and $\Delta G_{298}^{\neq }$(Aryl), respectively) listed in Table 1.

The calculated rotational barriers of the formyl group in N-benzhydrylformamide and its *o*-halogen derivatives are close to the analogous value for compound BHFA-NMe, differing from it slightly downwards, which may be due to the absence of a methyl substituent at the nitrogen atom in these molecules. On the whole, all the values of $\Delta G_{298}^{\neq }$(Formyl) found by the DFT method are very close to each other and practically independent of the presence and nature of the *ortho* substituent. As for the rotational barriers of the benzene ring with a halogen atom in the *ortho* position (or the unsubstituted phenyl group in BHFA and BHFA-NMe), they differ markedly for the compounds under study. The small-sized substituents—the *ortho*-fluorine atom or the *N*-methyl group—lead to a relatively weak increase in the value of $\Delta G_{298}^{\neq }$(Aryl) in comparison with compound BHFA (from 2.5 for BHFA to 3.1 and 5.8 kcal/mol for BHFA-NMe and BHFA-oF, respectively, Table 1), while chlorine, bromine, and iodine atoms increase the barrier to 8.7–9.8 kcal/mol. The closeness of the $\Delta G_{298}^{\neq }$(Aryl) values for BHFA-oBr and BHFA-oI is noteworthy, despite the noticeable difference in the van der Waals radii of the bromine and iodine atoms (1.95 and 2.15 Å, respectively [14]). This observation can be partially explained by differences in the electronegativity of halogens. The calculated ECP charges on F, Cl, Br, and I atoms are equal to −0.278, −0.103, −0.079, and +0.024, respectively, for the investigated *ortho*-substituted derivatives. Consequently, the iodine substituent in compound BHFA-oI has an attractive interaction with the electron-rich formamide moiety. Indeed, for the *ortho*-halogen derivatives under study, the calculated barriers $\Delta G_{298}^{\neq }$(Aryl) correspond to transition states in which the halogen atom is located near the formamide fragment. An example of such a transition state is shown in Figure 8. It should be mentioned again that it corresponds to the barrier penultimate in height on the energy profile found on scanning about the torsion angle N-CH-C_Ar_=C_Ar_ (Figure 7e), and the transition state with the maximum energy on this profile for BHFA-oI is characterized by a barrier of 10.7 kcal/mol and corresponds to the location of the iodine atom opposite the unsubstituted phenyl group.

To evaluate the relativistic effects that can be noticeable for bromo and iodo derivatives, we reoptimized the geometries of BHFA-oI, BHFA-oBr, BHFA-oCl, and the corresponding transition states using the ZORA approach [15]. The obtained rotational barriers are presented in Table 1. The $\Delta G_{298}^{\neq }$(Aryl) values are somewhat lower in the relativistic approximation than the barriers calculated with M06-2X functional. For the investigated iodo-, bromo-, and chloro-substituted benzhydrylformamides, the differences are equal to 0.57, 0.85, and 0.45 kcal/mol, respectively. Bearing in mind that for BHFA-oCl, this methodological difference has a mainly non-relativistic origin, we can conclude that for BHFA-oBr and BHFA-oI, the influence of relativistic effects on the barrier heights is not very significant.

It can be seen from Table 1 that the barriers for the rotation of the aryl group are much lower than for the formyl fragment in all benzhydrylformamides studied. Hence, at room temperature, the rotation of benzene rings must be very fast compared to the relaxation time of protons. In this regard, the presence of paired signals in the ^1^H NMR spectra is due only to slow *syn*-*anti* conformational transitions within the formamide group. The Gibbs energies $\Delta G_{298}^{\neq }$(*syn→anti*) of this transition (Table 1) obtained after optimization of the structures with *anti*-conformation taken from PES scans (Figure 6) confirm a higher stability of *syn*-conformers, which is more pronounced for BHFA and *ortho*-halogen substituted derivatives (1.4–1.7 kcal/mol). The reasons for the low value of $\Delta G_{298}^{\neq }$(*syn→anti*) obtained for BHFA-NMe were discussed in Section 2.2.1.

Internal rotation of the formyl group leads to a significant change in pyramidality of the nitrogen atom. According to our DFT calculations and the crystallographic data [6], the formamide moiety is fairly planar in *syn*- and *anti*-conformers, while in the corresponding transition states, the sum of valence angles at the nitrogen atom for *ortho*-halogen derivatives is about 325°; for BHFA and BHFA-NMe, this magnitude equals 328 and 335°, respectively, in agreement with earlier results on the influence of amide resonance on planarity of the nitrogen atom upon internal rotation [16,17,18].

## 3. Materials and Methods

### 3.1. The NMR Experiments

The ^1^H NMR spectra of compound BHFA-NMe synthesized according to the procedure described in [7] were recorded on a BRUKER AVANCE III HD instrument with an operating frequency of 400 MHz. DMSO-d_6_ was used as a solvent.

The study of coalescence of the methyl group signals was carried out by recording ^1^H NMR spectra at temperatures of 20, 35, 52, 58, 75, 95, 99, 110, and 120 °C.

The height of the internal rotational barrier ΔG^≠^ was determined by Equation (1) [8]:(1)$$\Delta G^{\neq }={\mathrm{RT}}_{c}[22.96+\ln\left(\frac{T_{c}}{\Delta \nu }\right)$$
where R is the universal gas constant;

T_c_—coalescence temperature, K;

Δν is the difference in frequency between the methyl group NMR signals of different conformers of compound BHFA-NMe, Hz.

### 3.2. Quantum Chemical Calculations

The initial conformational search in the benzhydrylformamide derivatives was performed by the method of molecular mechanics with the modified Dreiding force field [19] using the VConf program of the VeraChem software package (VeraChem LLC, Germantown, MD, USA). Ten conformations of each compound with the lowest energies were further optimized by DFT (see below), and the geometric structure with the lowest energy obtained after the optimization was chosen for further calculation of rotational barriers. Quantum chemical DFT calculations were performed using Gaussian 16 program [20] on a server (16 × 2.2 GHz CPU, 16 Gb RAM) operating under Ubuntu 16.04. The M06-2X functional [11] and the 6-311+G* basis set [21,22] were used on the calculations. For bromine and iodine atoms, the LANL2DZ basis with an effective core potential [23] was applied. The bulk effect of the solvent (dimethylsulfoxide) was taken into account within the IEFPCM solvation model [12].

One-dimensional scans of the PES with varying the torsion angles O=C-N-CH or N-CH-C_Ar_=C_Ar_ were carried out with a step of 20°. At each step, an optimization of all the other geometric parameters of the studied benzhydrylformamides was performed. To calculate a rotational barrier, we used the maximum penultimate height obtained by the scanning over the full period (360°) of each torsion angle. The barriers were calculated based on the characteristics of the corresponding transition states refined using the OPT = (TS, CALCFC, NOEIGEN) options of the Gaussian 16 program. For all the stationary points on the PES, the analysis of normal vibrations was carried out in order to establish the nature of the stationary point (a minimum or a transition state). The calculation results were visualized using GaussView 6.0 program [24].

The DFT-optimized low-energy conformers of BHFA-oCl, BHFA-oBr, BHFA-oI, and the corresponding transition states were reoptimized in the relativistic Hartree-Fock ZORA approximation [15] with ZORA-def2-SVP basis set and CPCM solvation model (dimethylsulfoxide). For clorine and bromine atoms, the triple-zeta ZORA-def2-TZVP basis was applied. For iodine atom, SARC-ZORA-TZVP basis set [25] was used. All the relativistic calculations were performed with the use of ORCA 5.0 quantum chemistry software [26].

The Gaussian and ORCA output files are available in the Appendix A.

## 4. Conclusions

New data on the conformational behavior of biologically important benzhydrylformamides were obtained. It was found that there are significant hindrances to the free rotation of the formyl group in these molecules, which stipulate the existence of N-benzhydrylformamides as mixtures of *syn*- and *anti*-conformers at ordinary temperatures. Since the properties of these conformers, including the ability to bind to receptors and enzymes, are different, the data obtained can be valuable for predicting the interaction of benzhydrylformamide derivatives with various biotargets.

## Data Availability

The data presented in this study are available in this article.

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
