# Peer review of "Rotational Barriers in N-Benzhydrylformamides: An NMR and DFT Study"

_molecules, 2023, doi:10.3390/molecules28020535_

Round 1

Reviewer 1 Report

The manuscript reports results of experimental and theoretical research on the internal rotation of aromatic and formyl fragments and estimation of the corresponding rotational barriers in N-benzhydrylformamide derivatives.

The manuscript brings interesting information though the authors should correct several flaws before publishing.

1.        There is a methodological inaccuracy in the article: the authors write about conformational analysis, but consider internal rotation only with respect to two bonds, while there is no data on the structure of the entire molecule of a particular compound in the text. Probably, the authors obtained these data as a result of DFT calculations, then they should be presented. In this case, the purpose of the research is to determine the value of the rotation barrier relative to a particular bond, but not a conformational analysis. In addition, in the abstract, the authors write about the conformations of the formyl group, but it is correct to discuss the position of the group in the conformers.

2.        The authors carried out the theoretical calculations of several compounds. It is not clear why they did not give such energy characteristics of the conformers as the relative energies and the Gibbs energies. Why the authors considered rotation only with respect to two bonds? What does "the best conformation" mean (lines 121, 182)? It is not clear how many preferred conformers were selected, what are their energy characteristics?

3.        The authors should justify the choice of DFT methods and calculation bases, as well as the model for taking into account the solvent effect. The calculation bases for various halogen derivatives differ, is it correct to compare the results obtained?

4.        The description of calculation procedures is repeated in sections 2.2.1., 2.2.2. and 3.2.

5.        Conclusions, lines 295-297: “It was found that there are significant hindrances to the free rotation of the formyl group in these molecules, which stipulate the existence of N-benzhydrylformamides as mixtures of syn- and anti-conformers at ordinary temperatures.” It is well known that ortho-substituents cause hindered rotation. In turn, the hindered rotation should lead to the predominance of a particular conformer.

6.        Did the authors evaluate the ratio of conformers of the studied compounds?

7.        The authors analyze the influence of the ortho-substituent in the aromatic ring in terms of size, was the electronegativity of halogens taken into account? There is no information about the effect of substituents on the electronic structure, the study of which was one of the aims of the research (lines 40-41).

8.        The images in Figure 2 should be merged.

9.        The order of references is broken in the manuscript: [8] is followed by [17] (page 2), [9]-[16] appear only on pages 10-11.

10.     Some references are formatted incorrectly ([4], [5], [7], [11]). Some abbreviations are wrong. There are no DOI in References. The vast majority of references relating to the objects of study refer to the authors' own works. The list of cited works should be expanded.

11.     The typos should be corrected, for example, Figure 3: how can the bond be directed to the group? The text should be cleaned in general.

Several misprints and grammatical errors can be found by more careful reading.

The results are worth to be published in Molecules; however, there are several points that the authors have to address before a final recommendation for acceptation can be made.

Author Response

  1. There is a methodological inaccuracy in the article: the authors write about conformational analysis, but consider internal rotation only with respect to two bonds, while there is no data on the structure of the entire molecule of a particular compound in the text. Probably, the authors obtained these data as a result of DFT calculations, then they should be presented. In this case, the purpose of the research is to determine the value of the rotation barrier relative to a particular bond, but not a conformational analysis. In addition, in the abstract, the authors write about the conformations of the formyl group, but it is correct to discuss the position of the group in the conformers.
    Thank you very much for the valuable comment. We have modified the title of the manuscript, the list of keywords, and made the correction in Abstract.
  2. The authors carried out the theoretical calculations of several compounds. It is not clear why they did not give such energy characteristics of the conformers as the relative energies and the Gibbs energies. Why the authors considered rotation only with respect to two bonds? What does "the bestconformation" mean (lines 121, 182)? It is not clear how many preferred conformers were selected, what are their energy characteristics?
    After the MM conformational search with VeraChem, ten low-energy conformers of each compound were selected and further optimized by the DFT method. The conformer Gibbs energies and the corresponding Gaussian output files were included in Supplementary Materials. The lowest-energy conformer (according to the DFT results) was then used for the relaxed scans with torsions about O=C-N-CH or N-CH-Car=Car bonds. These most important rotations define the orientations of amide group (syn or anti) and of sterically bulky o-substituted aromatic ring. The other two rotations (position of unsubstituted phenyl group and the mutual rotation of the benzhydryl and formamide moieties) are strongly correlated with these torsion angles. We clarified this issues in the manuscript and added Gaussian output files for the scan runs in Supplementary.
  3. The authors should justify the choice of DFT methods and calculation bases, as well as the model for taking into account the solvent effect. The calculation bases for various halogen derivatives differ, is it correct to compare the results obtained?
    Description of the choice of DFT method and solvation model was added in sections 2.2.1 and 2.2.2.
  4. The description of calculation procedures is repeated in sections 2.2.1., 2.2.2. and 3.2.
    We removed the repeated mentioning of the DFT details from sections 2.2.1 and 2.2.2.
  5. Conclusions, lines 295-297: “It was found that there are significant hindrances to the free rotation of the formyl group in these molecules, which stipulate the existence of N-benzhydrylformamides as mixtures of syn- and anti-conformers at ordinary temperatures.” It is well known that ortho-substituents cause hindered rotation. In turn, the hindered rotation should lead to the predominance of a particular conformer.
    We considered the hindered internal rotation of the amide group (i.e., the rotation of formyl group about the amide C-N bond) which is not obviously affected by the ortho-substituent.
  6. Did the authors evaluate the ratio of conformers of the studied compounds?
    The discussion of syn-anti interconversion energies and the ratio of conformers was added in Section 2.2.2.
  7. The authors analyze the influence of the ortho-substituent in the aromatic ring in terms of size, was the electronegativity of halogens taken into account? There is no information about the effect of substituents on the electronic structure, the study of which was one of the aims of the research (lines 40-41).
    The discussion on the influence of halogen electronegativity was added in Section 2.2.2.
  8. The images in Figure 2 should be merged.
    The images were merged.
  9. The order of references is broken in the manuscript: [8] is followed by [17] (page 2), [9]-[16] appear only on pages 10-11.
    The references were renumbered properly.
  10. Some references are formatted incorrectly ([4], [5], [7], [11]). Some abbreviations are wrong. There are no DOI in References. The vast majority of references relating to the objects of study refer to the authors' own works. The list of cited works should be expanded.
    The list of references was updated.
  11. The typos should be corrected, for example, Figure 3: how can the bond be directed to the group? The text should be cleaned in general.
    The word was changed to “connected”.

Several misprints and grammatical errors can be found by more careful reading.
The manuscript was checked thoroughly.

Reviewer 2 Report

The NMR part seems to be OK, however, the presentations of the DFT results must be improved.

1. About the formyl group rotation, please mention the energy differences between the syn- and anti-conformations  for BHFA and BHFA-oX, X=Cl, Br, I.

2. More detailed descriptions are necessary about the planarity of the amide group both at the ground state and transition state of the carbonyl orientations. The internal barrier of the C-N bond rotation are known to be influenced by the amide resonance structure, which also modify the geometry of the N atom in both the ground and transition states. The following references described this topic.

G. Fogarasi and P. G. Szalay, J. Phys. Chem. A 101, 1400-1408 (1997).

C. R. Kemnitz and M. J. Loewen, jacs 129, 2521-2528 (2007).

Y. Otani, et al., jacs 125, 15191-15199 (2003).

3. For the aryl group rotation, the ranges of lateral axes of Fig. 5(b) and Fig. 7(a-e) must be chosen the same to compare them more easily.

4. Because of the C2 symmetry of the phenyl group, Fig. 5(b) and  7(a) also have the C2 symmetry. Please assign the four maxima to the short atomic contacts by examining the geometrical data. Why two maxima in Fig. 7(a) become lower than those in Fig. 5(a).

5. The C2 symmetry of the phenyl group is lost by the ortho substitution. Now the potential has three maxima in Fig. 7(b-e). Please assign these three maxima to the short atomic contacts by examining the geometry data. 

Author Response

The NMR part seems to be OK, however, the presentations of the DFT results must be improved.

1. About the formyl group rotation, please mention the energy differences between the syn- and anti-conformations  for BHFA and BHFA-oX, X=Cl, Br, I.
We appended a column in Table 1 containing the energy differences between the syn- and anti-conformations and added an explanation in the text.

2. More detailed descriptions are necessary about the planarity of the amide group both at the ground state and transition state of the carbonyl orientations. The internal barrier of the C-N bond rotation are known to be influenced by the amide resonance structure, which also modify the geometry of the N atom in both the ground and transition states. The following references described this topic.

Fogarasi and P. G. Szalay, J. Phys. Chem. A 101, 1400-1408 (1997).

R. Kemnitz and M. J. Loewen, jacs 129, 2521-2528 (2007).

Otani, et al., jacs 125, 15191-15199 (2003).

The description of planarity (pyramidality) of amide group with the literature references was added in Section 2.2.2.

3. For the aryl group rotation, the ranges of lateral axes of Fig. 5(b) and Fig. 7(a-e) must be chosen the same to compare them more easily.
This is quite difficult, as these images were generated by GaussView software directly from output of the calculations which had already been performed. However, we included Gaussian output files of the PES scans in Supplementary Materials so that everyone could open and analyze them in terms of energy variation with conformational changes.

3. Because of the C2 symmetry of the phenyl group, Fig. 5(b) and  7(a) also have the C2 symmetry. Please assign the four maxima to the short atomic contacts by examining the geometrical data. Why two maxima in Fig. 7(a) become lower than those in Fig. 5(a).
The explanation was given in caption to Figure 7.

4. The C2 symmetry of the phenyl group is lost by the ortho substitution. Now the potential has three maxima in Fig. 7(b-e). Please assign these three maxima to the short atomic contacts by examining the geometry data. 
The assignments were described in caption to Figure 7.

Reviewer 3 Report

Notwithstanding this article neither introduces new physical insights nor develops new methodologies, it is surely very well written, comprehensively done and represents a saliant example of integrity of dynamic NMR experimental and quantum chemical study of the conformational behavior of N-benzhydrylformamides. I will recommend this manuscript for publishing in Molecules should the authors comply with my requests and answer my questions.

1. Compounds BHFA-oBr and BHFA-oI bear heavy atoms, bromine and iodine. Is it not unreasonable to perform the geometry optimizations without taking care of relativistic effects? The authors could have carried out the optimizations with taking into account the scalar (or spin-free) relativistic effects at least. In particular, the lengths of the C-I or C-Br bonds may significantly vary depending on whether one resorts to the relativistic level of theory when performing the optimization procedure or not.  The same question arises when one calculates the energies of bromine- and iodine-containing compounds. What portions of the total molecular energies of such compounds occurs to be fallen out due to unaccounted relativistic corrections?  What errors may this cause when calculating the rotational barriers? I understand that it is impossible to recalculate all at the relativistic level of theory right now, though, I strongly recommend to speculate on this topic in the text leaning on the available literature sources.

2. The experiment was said to be carried out in DMSO; hence, the quantum chemical calculations were performed using the IEF-PCM model with DMSO. However, purely continuum description like that within the IEF-PCM model might fail in the case of DMSO, for this is known to be a strong polar solvent (dielectric constant = 46.826) engaging into the specific solute-solvent interactions. This issue deserves some more portion of attention in the manuscript. The authors should clarify why they did not use supermolecule approach in such an intricate case as this. Or, mayhap, they could not perform their NMR experiments in the other less polar solvents. Why? All this requires a clarification. 

Author Response

Notwithstanding this article neither introduces new physical insights nor develops new methodologies, it is surely very well written, comprehensively done and represents a saliant example of integrity of dynamic NMR experimental and quantum chemical study of the conformational behavior of N-benzhydrylformamides. I will recommend this manuscript for publishing in Molecules should the authors comply with my requests and answer my questions.

  1. Compounds BHFA-oBr and BHFA-oI bear heavy atoms, bromine and iodine. Is it not unreasonable to perform the geometry optimizations without taking care of relativistic effects? The authors could have carried out the optimizations with taking into account the scalar (or spin-free) relativistic effects at least. In particular, the lengths of the C-I or C-Br bonds may significantly vary depending on whether one resorts to the relativistic level of theory when performing the optimization procedure or not.  The same question arises when one calculates the energies of bromine- and iodine-containing compounds. What portions of the total molecular energies of such compounds occurs to be fallen out due to unaccounted relativistic corrections?  What errors may this cause when calculating the rotational barriers? I understand that it is impossible to recalculate all at the relativistic level of theory right now, though, I strongly recommend to speculate on this topic in the text leaning on the available literature sources.
    We carried out the relativistic calculations for Cl, Br, and I substituted derivatives and discussed the results in the manuscript.
  2. The experiment was said to be carried out in DMSO; hence, the quantum chemical calculations were performed using the IEF-PCM model with DMSO. However, purely continuum description like that within the IEF-PCM model might fail in the case of DMSO, for this is known to be a strong polar solvent (dielectric constant = 46.826) engaging into the specific solute-solvent interactions. This issue deserves some more portion of attention in the manuscript. The authors should clarify why they did not use supermolecule approach in such an intricate case as this. Or, mayhap, they could not perform their NMR experiments in the other less polar solvents. Why? All this requires a clarification. 
    We clarified these issues in the manuscript.

Round 2

Reviewer 1 Report

After revision, the manuscript can be published.